# Study on the Hemispheric Asymmetry of Thermospheric Density Based on In-Situ Measurements from APOD Satellite

**Jiangzhao Ai [1,2,*], Yongping Li [1,2], Xianguo Zhang [1,2], Chao Xiao [3], Guangming Chen [4], Xiaoliang Zheng [1,2] and Zhiliang Zhang [1,2]**

[1] National Space Science Center, Chinese Academy of Sciences, Beijing 100190, China
[2] Beijing Key Laboratory of Space Environment Exploration, Beijing 100190, China
[3] Institute of Space Sciences, School of Space Science and Physics, Shandong University, Weihai 264200, China
[4] Science and Technology on Aerospace Flight Dynamics Laboratory, Beijing Aerospace Control Center, Beijing 100094, China
[*] Correspondence: aijiangzhao@nssc.ac.cn

**Abstract:** In this article, high spatiotemporal resolution data obtained by the atmospheric density detector carried by China's APOD satellite are used to study the hemispheric asymmetry of thermospheric density. A detailed analysis is first performed on the dual magnetic storm event that occurred near the autumnal equinox on 8 September 2017. The results show that the enhancement ratio of atmospheric density in the southern polar region (SPR) on the duskside was approximately 1.33–1.65 times that of the northern polar region (NPR), demonstrating a strong hemispheric asymmetry of thermospheric atmospheric density response during the magnetic storm. However, the asymmetry response was smaller on the dawnside, suggesting that the hemispheric density response asymmetry is related to local time (LT). The energy injection in high-latitude regions increases local atmospheric density and forms traveling atmospheric disturbances (TADs). TADs can propagate to low-latitude regions over several hours and affect the global distribution of thermospheric atmospheric density. Similarly, the geomagnetic index fitting slope of SPR relative density difference is greater than that of NPR. The SuperDARN convection pattern indicates that the plasma convection velocity of SPR is significantly greater than that of NPR, indicating that joule heating caused by neutral friction of ions in the Southern Hemisphere may be stronger. Subsequently, an analysis of annual solar activity and seasons was carried out on the thermospheric NPR, SPR atmospheric density, and their differences from December 2015 to December 2020. The results show that thermospheric atmospheric density decreases overall as the number of sunspots decreases. The differences between the NPR and SPR atmospheric densities in the thermosphere exhibits a noticeable annual periodicity. The NPR and SPR atmospheric densities appear to have different distribution characteristics in different seasons. The NPR density peak is mainly in March or April. In particular, the "double-peak" phenomenon occurred in 2017, with peaks in March and September, while the most obvious feature of SPR atmospheric density is that its minimum value occurs in the summer months of June and July. This paper reveals the annual, seasonal, and magnetic storm response characteristics of the hemispheric asymmetry of thermospheric atmospheric density, which has significant implications for the study of multilayer energy coupling of the magnetosphere–ionosphere–thermosphere.

**Keywords:** atmospheric density; thermosphere; hemispheric asymmetry; TADs; plasma convection

## 1. Introduction

The thermosphere is a layer of the Earth's atmosphere that extends from 80 km to 1000 km above the surface. The primary sources of energy input to the thermosphere include solar extreme ultraviolet (EUV) radiation, auroral particle precipitation, and joule heating in high-latitude regions, which strongly influence the thermosphere's structure and dynamics. Quantifying the degree of hemispheric asymmetric response of these main

energy input sources to the neutral density and composition of the thermosphere has always been an important research topic. Due to the different solar irradiation received in seasons, thermosphere mass density and composition can also show corresponding seasonal variations [1,2]. For instance, Qian et al. [3] combined Thermosphere–Ionosphere–Electrodynamics General Circulation Model (TIE-GCM) simulations and TIMED/GUVI observations to show that the neutral density and composition of the thermosphere presents strong seasonal variations, with maximums near the vernal equinox, primary minimums in the Northern Hemisphere summer, and secondary minimums in the Southern Hemisphere summer. Furthermore, high-latitude joule heating, which is a source of energy input to the thermosphere, is affected by solar wind conditions as well as geomagnetic activity [4,5]. During geomagnetic storms, the enhanced high-latitude joule heating causes thermal expansion of the thermosphere, which in turn causes disturbances in mass density. For example, Sutton et al. [6] analyzed accelerometer data from the CHAMP satellite at an altitude of around 410 km during the geomagnetic storm on 29 October to 1 November 2003, and found that the atmospheric density increased by 200–300%. Pham et al. [7] used accelerometer data from the CHAMP and GRACE satellites during the 24–25 August 2005 geomagnetic storm to detect dramatic perturbations of atmospheric density in the global thermosphere. Particle precipitation in the polar cusp also plays a certain role in the rise of atmospheric density in the thermosphere. The results of Liu et al. [8] showed that atmospheric density at high latitude is highly structured. Later, Zhang et al. [9] used the simulation results of the coupled magnetosphere–ionosphere–thermosphere (CMIT) model to explain that high-latitude density structure is related to soft electron precipitation. Some studies also indicated that the small-scale field-aligned current in the polar cusp region can cause an increase in the atmospheric density of the thermosphere [10,11]. In addition, joule heating and particle precipitation in the high-latitude region generate upwelling air [10] that changes the density ratio of oxygen to nitrogen ($\sum O/N2$) in the thermosphere [12,13].

Previous studies have shown that there is hemispheric asymmetry in the response of thermosphere mass density to geomagnetic storms [14–16]. The analysis results of the magnetic storm event on 21–22 November 2003 showed that there was a significant difference in the density response ratio of north and south geomagnetic latitude 72° [14]. It has also been found that differences in hemispheric response in thermosphere density are related to local time (LT). Li and Lei [17] analyzed the thermospheric variations during the geomagnetic storm periods that occurred near the autumnal equinox in October 2016 and September 2017, respectively, using the precise orbit determination derived from GRACE, Swarm-A, and Swarm-B satellite measurements of atmospheric mass density. The results showed that the density perturbation around 16 LT of the Southern Hemisphere was stronger than the Northern Hemisphere. However, the density enhancements in the Northern and Southern hemispheres is almost symmetrical around 08:00/20:00 LTs. This difference in LT-related thermosphere density hemispheric response may be related to the asymmetry of joule heating LT distribution at high latitudes, but there are currently insufficient observational data to prove it.

Therefore, we need to discuss the external source asymmetry that leads to differences in thermosphere density response. First, the asymmetry of solar EUV radiation may result in an asymmetry in ionospheric conductivity, which further affects plasma convection [18,19] and field-aligned current [10]. In the past, many studies have analyzed plasma convection derived from satellite in situ measurements, such as DMSP and Super Dual Auroral Radar Network (SuperDARN) data. Koustov et al. [20] exhibited an asymmetric two-cell convection pattern with a large dusk cell by combining SuperDARN and swarm data. Second, auroral particle precipitation also has hemispheric asymmetry. By analyzing data from the Global Ultraviolet Imager aboard the Thermosphere–Ionosphere–Mesosphere Energetics and Dynamics satellite from 2002 to 2007, Liou and Mitchell [21] found that the dayside auroral energy flux showed a small north–south asymmetry. Moreover, north–south geomagnetic field asymmetry may cause differences in plasma convection, such as large tilt angle in the Southern Hemisphere, resulting in greater plasma drift velocity compared to

the Northern Hemisphere [14,22]. Therefore, quantifying the difference in thermosphere density response caused by external input sources is of great significance for understanding the coupling between the solar wind–magnetosphere–ionosphere–thermosphere system. In this study, high spatial and temporal resolution data of thermospheric density measured in situ by the atmospheric density detector aboard China's APOD satellite are used to investigate the hemispheric asymmetry characteristics of thermospheric density. Section 2 provides a brief introduction to the APOD satellite and its payload. In Section 3, the geomagnetic response of thermospheric density in both hemispheres during a dual magnetic storm event that occurred near the autumnal equinox on 8 September 2017 is studied, as the magnetic storm occurred near the equinox and is therefore easier to analyze. Then, seasonal and annual solar activity phenomena of northern polar region (NPR—latitude above 60° N) and southern polar region (SPR—latitude above 60° S) thermospheric densities, as well as their differences, are explored from December 2015 to December 2020.

## 2. Satellite Description

The APOD (Atmospheric Density Detection and Precise Orbit Determination) satellite was launched on 20 September 2015, and entered a nearly circular orbit with an inclination of 97.4° and an altitude of approximately 460 km on 27 October 2015. It is the first Chinese micro–nanosatellite platform with the primary scientific goals of detecting thermosphere atmospheric density and precise satellite orbit determination [23]. The high-inclination orbit of APOD is characterized by its ability to cover a large latitude range during the ascending and descending phases. As shown in Figure 1b, the ascending and descending branches of the APOD orbit are on the duskside and dawnside, respectively. In addition, the orbital precession of the satellite is very small: local time of the descending node drifted from 6.2 h in December 2015 to 9.0 h in December 2020, with a change of about 3 h during the period. The main instruments carried by APOD include the Atmospheric Density Detector (ADD) and a GNSS receiver. The atmospheric density data used in this paper are derived from the in situ measurements of the ADD, which directly measures pressure and temperature in the sampling chamber with sampling rate of 1 Hz and measurement error of density about 10% [24]. The high temporal and spatial resolution of the ADD provides a good opportunity to analyze the hemispheric response differences in thermospheric density during magnetic storms, as well as the detailed wave characteristics.

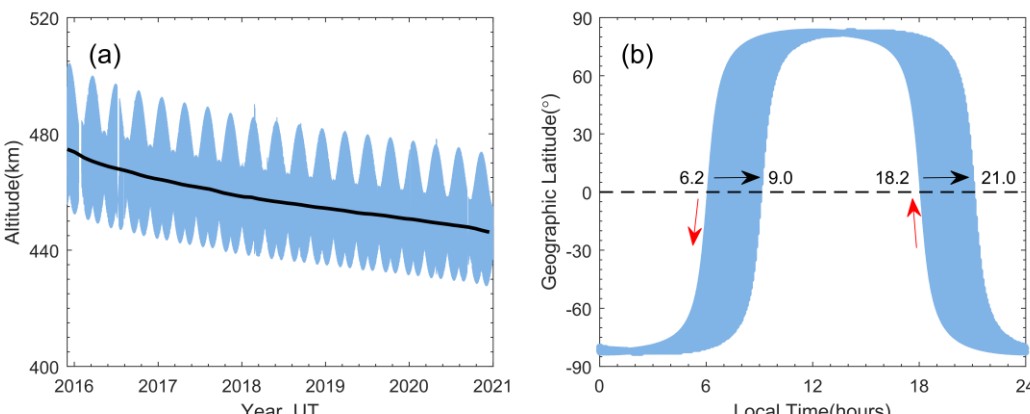

**Figure 1.** Changes in the spatial position of the APOD satellite from December 2015 to December 2022: (**a**) altitude, with the solid black line indicating the monthly mean altitude; (**b**) latitude and local time (LT) relationship, with the red arrow indicating the direction of the satellite's motion on a single orbit, and the black arrow indicating the direction of the satellite's LT precession.

From Figure 1a, it can be seen that the monthly mean altitude of the APOD satellite orbit decreased from an initial ~475 km in December 2015 to ~445 km in December 2020. Due to the existence of a certain eccentricity in the orbit, the altitude difference between the apogee and perigee can reach about 30 km. Some past research on thermospheric neutral

atmospheric density usually normalized the original detection value of atmospheric density to a certain altitude using the NRLMSISE-00 model, so as to reduce the error caused by the orbital altitude change [17,25]. However, when the diffusion balance condition is broken during a magnetic storm, the normalization of the density may lead to greater errors [8,26]. Therefore, in this study, the raw atmospheric density measurements were not normalized, and the relative differences in density are analyzed to minimize the influence of density on altitude dependence.

## 3. Results

### 3.1. Analysis of the 8 September 2017 Geomagnetic Event

3.1.1. Geomagnetic Conditions during the Storm on 8 September 2017

Figure 2 shows the variations in solar wind and geomagnetic disturbance parameters from 7 to 10 September 2017. The magnetic storm event on 8 September was a dual storm caused by the interaction of coronal mass ejection (CME) and the magnetosphere [27]. As the event occurred near the autumnal equinox, it was conducive to studying the hemispheric response differences of density during the magnetic storm period. From Figure 2a,b, it can be seen that at around 00:00 UT on 8 September, there was a sudden increase in solar wind velocity and dynamic pressure (the wind speed and dynamic pressure increased from 473 km/s and 1.5 nPa to 690 km/s and 4.35 nPa, respectively), while the interplanetary magnetic field (IMF) component Bz decreased to $-24.2$ nT, and both AE and Ap indices quickly increased to 1157 nT and 207 nT, respectively. The first magnetic storm reached its peak, and the Dst index decreased from an initial value of 13 nT to $-122$ nT. Subsequently, the solar wind velocity continued to rise slightly, and the dynamic pressure remained oscillating back and forth before rapidly increasing to 7.93 nPa at 10:00 UT. Then, Bz began to turn from north to south, providing the basic conditions for the injection of solar wind energy and momentum into the Earth's space. The corresponding AE and Ap indices quickly increased to 1442 nT and 236 nT. The first magnetic storm had not yet returned to its predisturbance state (at 10:30 UT, the Dst index was $-44$ nT), and then the second magnetic storm began to approach, with the minimum value of Dst index during the main phase being $-109$ nT. Therefore, based on the effects of the first magnetic storm, the impact of the second storm on the thermosphere might be greater.

3.1.2. Mass Density Behavior from 7 to 10 September 2017

Figure 3 displays the hemispheric response differences of the thermospheric density measured by the ADD instrument during the magnetic storm period from 7 to 10 September 2017. The top row of Figure 3 shows the density changes on the dusk side around 19:20 LT. From Figure 3a, it can be seen that there is a neutral density trough in the range of $40^\circ$ N to $60^\circ$ N around 04:00 UT, before the magnetic storm on 7 September, which is similar to the mid-latitude trough phenomenon in the Northern Hemisphere ionospheric electron density [28]. Figure 3b,c show that after the arrival of the first magnetic storm, the density values near $80^\circ$ N in the Arctic increased to about $1.32 \times 10^{-12}$ kg/m$^3$, while the density values around $80^\circ$ S in the Southern Hemisphere changed from $0.48 \times 10^{-12}$ kg/m$^3$ during quiet periods to $0.87 \times 10^{-12}$ kg/m$^3$. Moreover, compared to the first magnetic storm, the density perturbation range and uplift amplitude caused by the second storm were larger, and the disturbance lasted longer, approximately 15 h. In addition, the density value of SPR slightly increased during 09:00–12:00 UT on the 7th, as shown in Figure 3c, which corresponds to the first peak of the AE index in Figure 2d. The density perturbation responses of NPR and SPR have a lag of several hours with respect to the Ap and Dst geomagnetic indices. Similar to the presentation of the density on the duskside, the second row in Figure 3 shows the response of the density to the magnetic storm on the dawnside (~07:20 LT). Figure 3e shows that during the first magnetic storm, the density increased to about $1.38 \times 10^{-12}$ kg/m$^3$ near 80ºN in the Arctic, and the density disturbance range was in the region of $65^\circ$–$80^\circ$ N. Unlike the propagation of density disturbances on the duskside, the propagation of density disturbances on the dawnside NPR seems to be

strongly "resisted," and the peak density disturbance decreases rapidly in the latitude direction, as if it is "confined" in the high-latitude region. In addition, the density at low and mid-latitudes increases significantly, as shown in Figure 3a,e. This is mainly caused by traveling atmospheric disturbances (TADs) and will be discussed in detail. To reduce the density errors caused by changes in orbital altitude, the density during the quiet period was taken as a reference value, and the deviation of the density during the disturbance period relative to the reference value was calculated to quantify the differences in density response between the two hemispheres during the magnetic storm, as shown in Figure 3d,h. The results show that the relative density difference of the SPR is higher than that of the NPR, indicating a difference in density response in the high-latitude regions of the Northern and Southern hemispheres. The enhancement ratio of atmospheric density in the SPR is about 1.33–1.65 times that of the NPR on the duskside, and the north–south response difference on the dawnside is smaller than on the duskside. During this period, the mean relative density difference of the NPR and SPR on the duskside and dawnside was −9.01% and −5.59%, respectively. At the same time, the dependence of the north–south high-latitude density response difference on LT is also reflected.

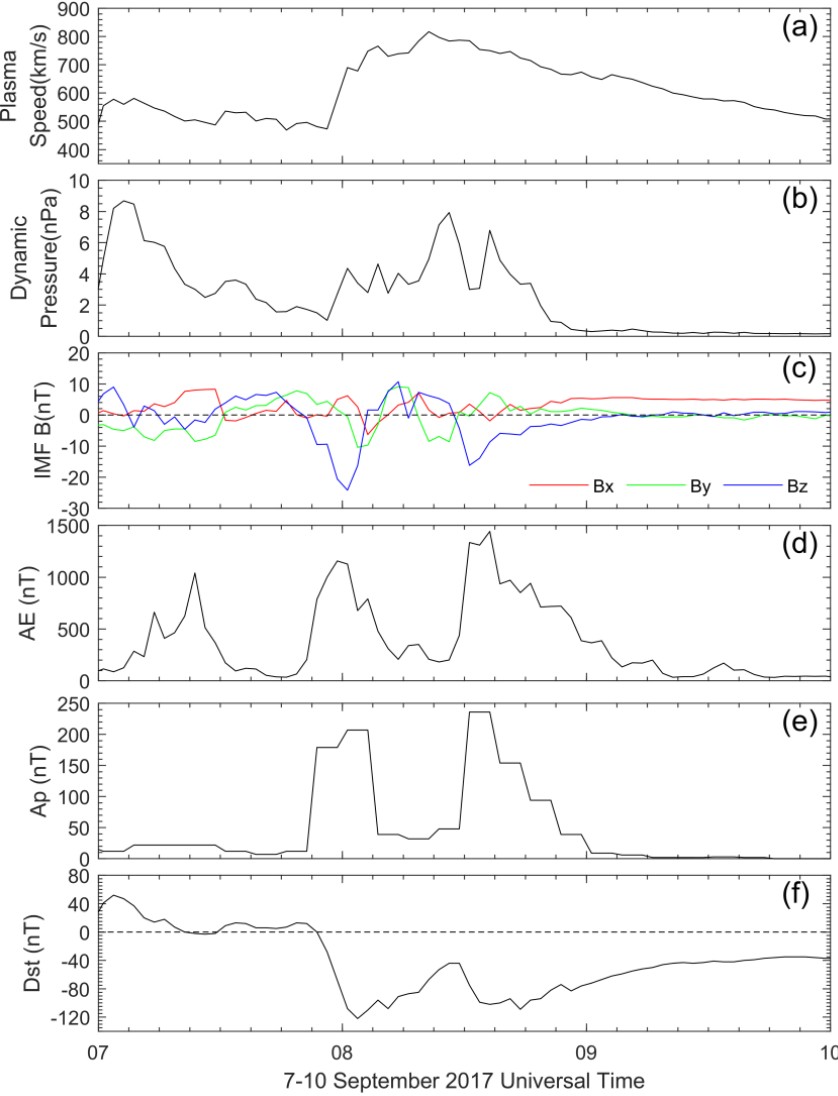

**Figure 2.** Variations in solar wind and geomagnetic parameters from 7 to 10 September 2017. From top to bottom are (**a**) solar wind velocity, (**b**) solar wind dynamic pressure, (**c**) interplanetary magnetic field Bx, By and Bz, (**d**) AE, (**e**) Ap, and (**f**) Dst.

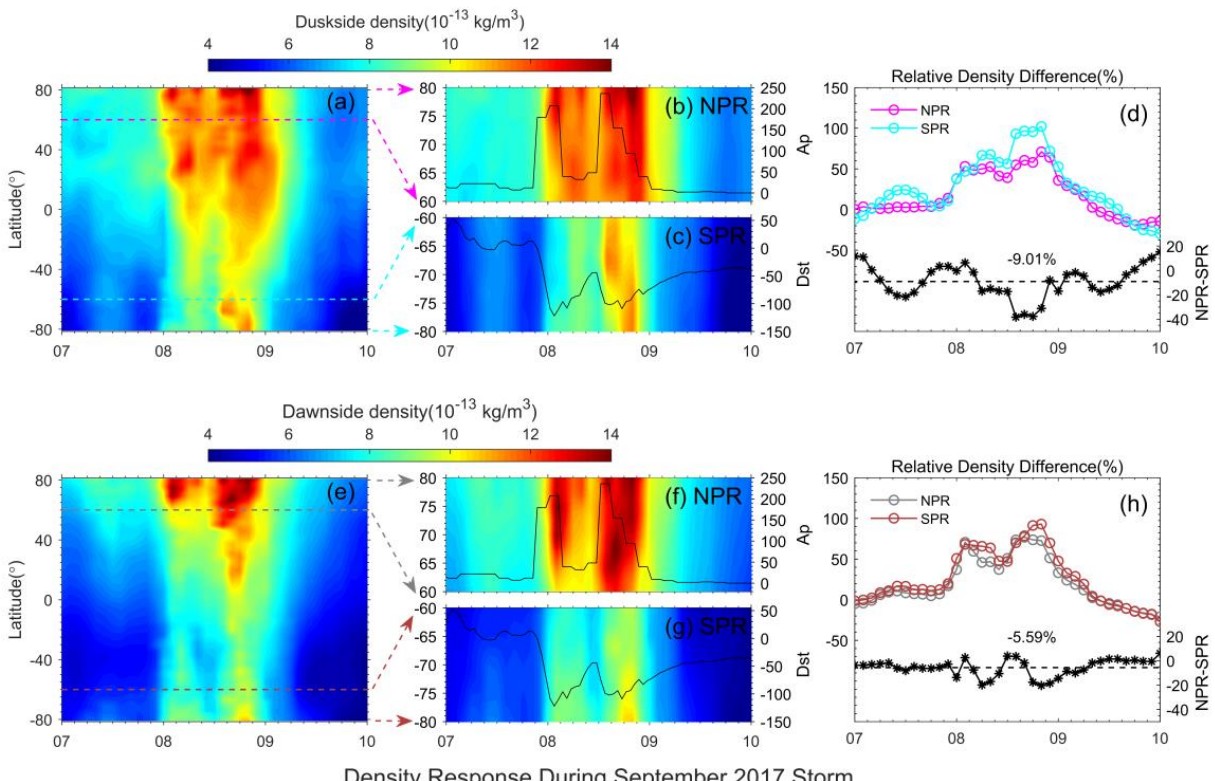

**Figure 3.** Thermospheric density response during the geomagnetic storm from 7 to 10 September 2017. Panels (**a**,**e**) display the global distribution of thermospheric density variations with UT and latitude on the dusk- and dawnside, respectively. Panels (**b**,**c**) show the density latitudinal profiles for the northern polar region (NPR, above 60° N) and southern polar region (SPR, above 60° S) on the duskside, with corresponding black lines indicating the Ap and Dst geomagnetic indices. Similarly, panels (**f**,**g**) display the density latitudinal profiles for the NPR and SPR on the dawnside. Panels (**d**,**h**) show the relative density difference on the duskside and dawnside, respectively, with the colored lines indicating the relative density difference for the NPR and SPR and the black line indicating the difference in relative density difference between the NPR and SPR, with the mean difference also labeled on the plot.

As analyzed earlier, during periods of strong geomagnetic disturbances, a large amount of energy is deposited from the magnetosphere to the high-latitude thermosphere, resulting in a sudden increase in atmospheric density at high latitudes. This atmospheric disturbance, generated by joule heating, can propagate or diffuse from high-latitude regions to lower latitudes, the equator, and even the opposite hemisphere, and is referred to as TADs by many researchers [7,29,30]. Figure 4 shows the changes in the atmospheric density distribution on the morning side of the APOD satellite from the 5th to the 10th. It should be noted that the atmospheric density of the Northern Hemisphere is slightly higher than that of the Southern Hemisphere during the quiet period (black dashed line). This is because the Northern Hemisphere is in summer during this period and thus warmer than the Southern Hemisphere. When a magnetic storm occurs (magenta dotted line), the overall global atmospheric density rises by nearly 50%, and there is a strong disturbance in the density of the Northern Hemisphere above 70° N. At the same time, TAD propagation is clearly observed in the mid-latitude regions of the Southern Hemisphere. As shown in Figure 4b, TADs began to disappear after the first geomagnetic storm. With the arrival of the second storm, disturbances in the high-latitude atmosphere of the Northern Hemisphere formed TADs, which continued to strengthen and propagate to the 30° N region as energy continued to be injected. At the same time, TADs formed in the Southern Hemisphere near 65° S and began to transport towards lower latitudes. After 2 h, at 16:00

UT, TADs in both hemispheres propagated to latitudes near 20° (as shown by the solid green line) and continued to propagate towards the equator, interfered with each other, and then spread to the opposite hemisphere, as shown by the solid blue line. Therefore, energy injection in high-latitude regions increases local atmospheric density and forms TADs during geomagnetic storms. TADs can propagate to low-latitude regions within hours and affect the all-latitude distribution of thermospheric atmospheric density.

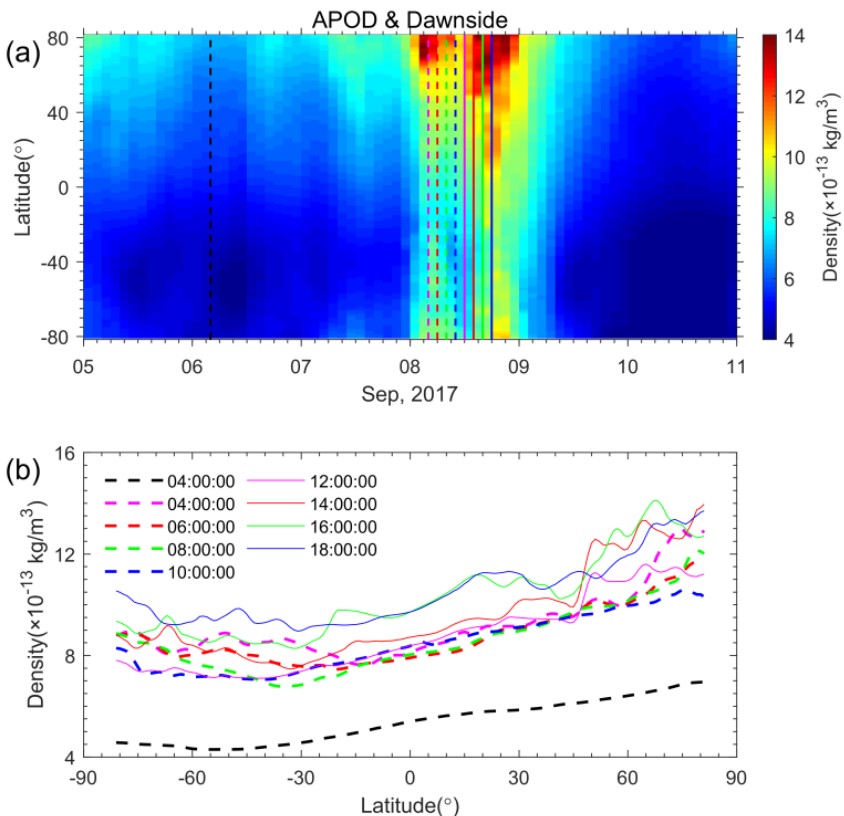

**Figure 4.** Propagation process of TADs on the dawnside during magnetic storm. Panel (**a**) displays the global distribution of thermospheric density variations with UT (from 5 to 11 September 2017) and latitude on the dawnside; Panel (**b**) shows the variation of density at different UTs with latitude during magnetic storm.

In order to more clearly and quantitatively analyze the asymmetry of the Northern and Southern hemisphere responses of the thermospheric atmospheric density caused by this magnetic storm event, a linear fitting discussion was carried out using the Dst/Ap indices and the relative density difference of NPR and SPR, as shown in Figure 5. The magenta and green dots in Figure 5a represent the relative density difference of NPR and SPR, respectively, as well as the linear regression lines fitted with Dst. Similarly, the blue and red dots in Figure 5b represent the relative density difference of NPR and SPR, respectively, as well as the linear regression lines fitted with Ap. The corresponding regression slopes are also indicated in each subfigure, and the data points in all subfigures include both the dawn and dusk sectors. As shown in the figure, the regression slopes of the NPR relative density difference with respect to the Dst and Ap indices are 0.467 and 0.247, respectively, while the corresponding SPR relative density difference regression slopes are 0.545 and 0.269. Therefore, the SPR relative density difference has a higher regression slope than the NPR, indicating a stronger response of Southern Hemisphere high-latitude atmospheric density to the geomagnetic storm event. This is consistent with the previously discussed difference in the geomagnetic storm response of atmospheric density between the two hemispheres.

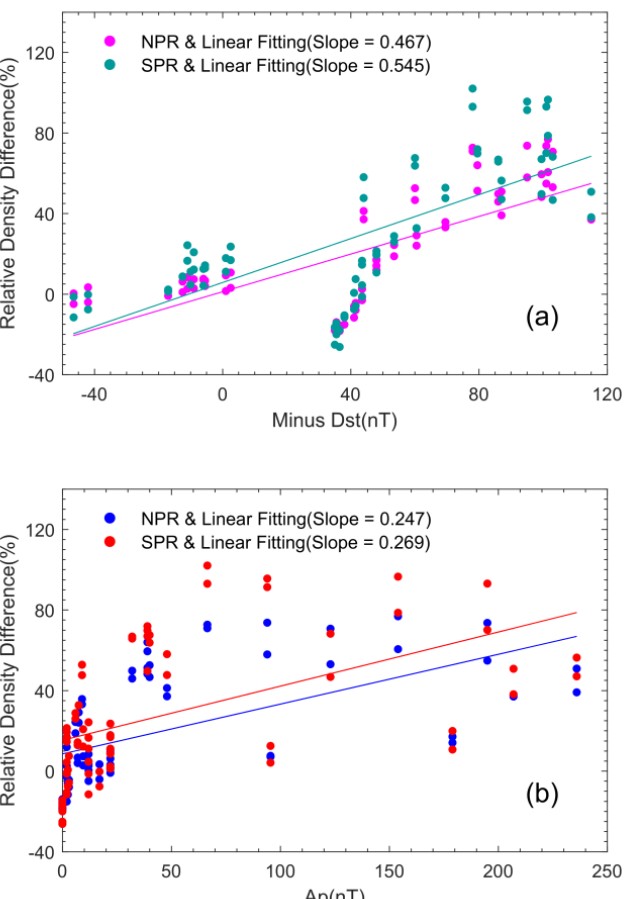

**Figure 5.** The relationship between the relative density difference of NPR and SPR versus the negative Dst index (**a**) and Ap index (**b**).

### 3.1.3. Explanation of Density Response Asymmetry

From the previous analysis of the magnetic storm event of 8 September 2017, the density response in the SPR is higher than the NPR. In this section, we explain this asymmetry phenomenon through plasma convection in the Northern and Southern hemispheres. Because the event occurs near the autumnal equinox, the zenith angle of the two hemispheres relative to the sun are similar, so the difference in conductivity between the NPR and SPR is small. SuperDARN, an international terrestrial high-frequency radar network, measures the line-of-sight (LOS) component of plasma E × B drift of ionosphere at high latitudes in both hemispheres. The plasma convection data used in this section are fitted velocities. The following steps are required to perform from LOS velocity to fitted velocity. First, the LOS velocity data are smoothened by median filtering of the measurements of several adjacent beams and gates and three consecutive sweeps. The obtained velocities are then sorted and filled with a uniform magnetic latitude and longitude grid. Finally, the obtained grid velocity is fitted to a plasma flow statistical model with solar wind and IMF conditions, the Earth's magnetic field, and dipole tilt angle as input parameters [20,31].

Figure 6 shows the plasma flow pattern at high latitudes measured by the SuperDARN that consists of ten radars in the Northern Hemisphere and six radars in the Southern Hemisphere. The convection pattern is represented by a flow velocity vector (colored vector) at each grid point (corresponding color point) and a line with equal electric potential (blue and red closed lines). The pattern uses magnetic latitude (MLAT)–magnetic local time (MLT) coordinates at 2 min intervals. The 17:52–17:54 UT corresponds to the moment when the density response asymmetry of the NPR and SPR is large in Figure 3. The data in Figure 6a,b show the SuperDARN convection pattern of an asymmetric two-cell with larger dusk cell, which is consistent with previous analysis that the density response asymmetry

on the duskside greater than dawnside. In addition, the plasma convection velocity of the SPR is significantly greater than that of the NPR from Figure 6a,b. The convection velocity of 11–12 MLT and MLAT around 73° in the Northern Hemisphere is about 300 m/s, while the Southern Hemisphere can reach 600–800 m/s. This north–south convection asymmetry indicates stronger joule heating caused by ion-neutral friction in the Southern Hemisphere, resulting in density response asymmetry in the NPR and SPR. In addition to this UT range, plasma convection patterns of the Southern and Northern hemispheres during the main phase of the second magnetic storm are analyzed. The results render that plasma convection velocity in the high-latitude region of the Southern Hemisphere is larger than in the Northern Hemisphere during the main phase. This means that the energy injected into the polar regions of the Southern Hemisphere may be higher than in the Northern Hemisphere.

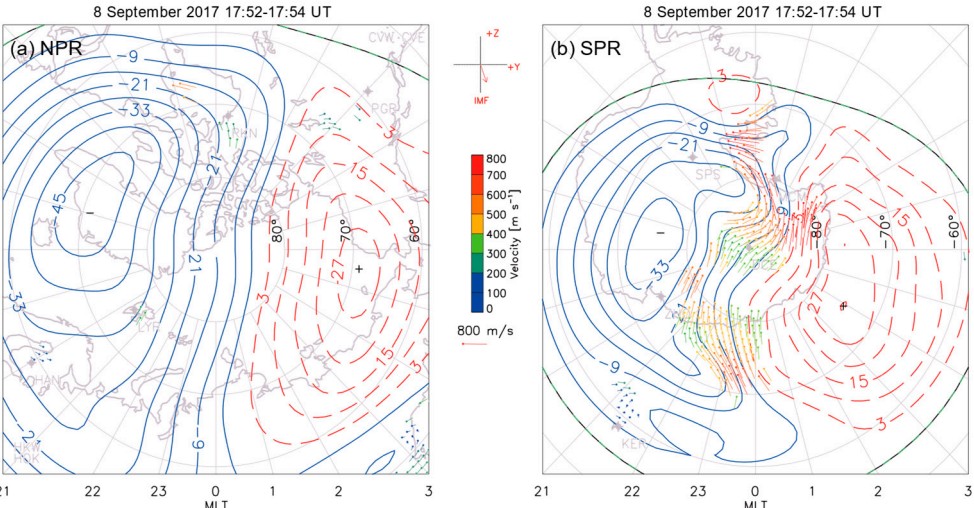

**Figure 6.** The two-minute SuperDARN convection pattern with E × B plasma flow pattern, characterized by colored vectors at individual grid measurement points and by equal potential lines (blue and red closed lines) plotted with 6 kV steps. (**a**): NPR; (**b**): SPR. IMF and flow velocity reference vector also marked in the figure.

### 3.2. Mid- and Long-Term Observations of North-South Asymmetry

The previous section previously analyzed the differences in atmospheric density response between the NPR and SPR in a single geomagnetic storm event. This section discusses the mid- to long-term variations in north–south asymmetry in thermospheric high-latitude atmospheric density. Figure 7 shows the time evolution of the thermospheric NPR, SPR, global region (GR) atmospheric density, and NPR-SPR atmospheric density difference, as well as the corresponding geomagnetic indices Dst and Ap, observed by the APOD satellite from December 2015 to December 2020.The results indicate that the thermospheric atmospheric density exhibits distinct features due to annual solar activity, seasonal variation, and disturbance under different geomagnetic activity conditions. The thermospheric GR atmospheric density decreases as the sunspot number decreases, as shown in Figure 7c,d. Figure 7e,f show that NPR and SPR atmospheric density have seasonal characteristics. NPR atmospheric density is lower in winter than in other seasons, while SPR atmospheric density is lower in summer. The thermospheric NPR and SPR atmospheric density also reflect sudden increases in short-term ranges due to geomagnetic disturbances. Figure 7g shows the percentage difference in atmospheric density between the NPR and SPR, calculated as $(\rho_{NPR} - \rho_{SPR})/(\rho_{NPR} + \rho_{SPR})$. The results indicate that the differences in atmospheric density between the high-latitude regions of the two hemispheres exhibit a significant annual periodicity. Therefore, thermospheric atmospheric density parameters are affected by annual solar activity and different seasons.

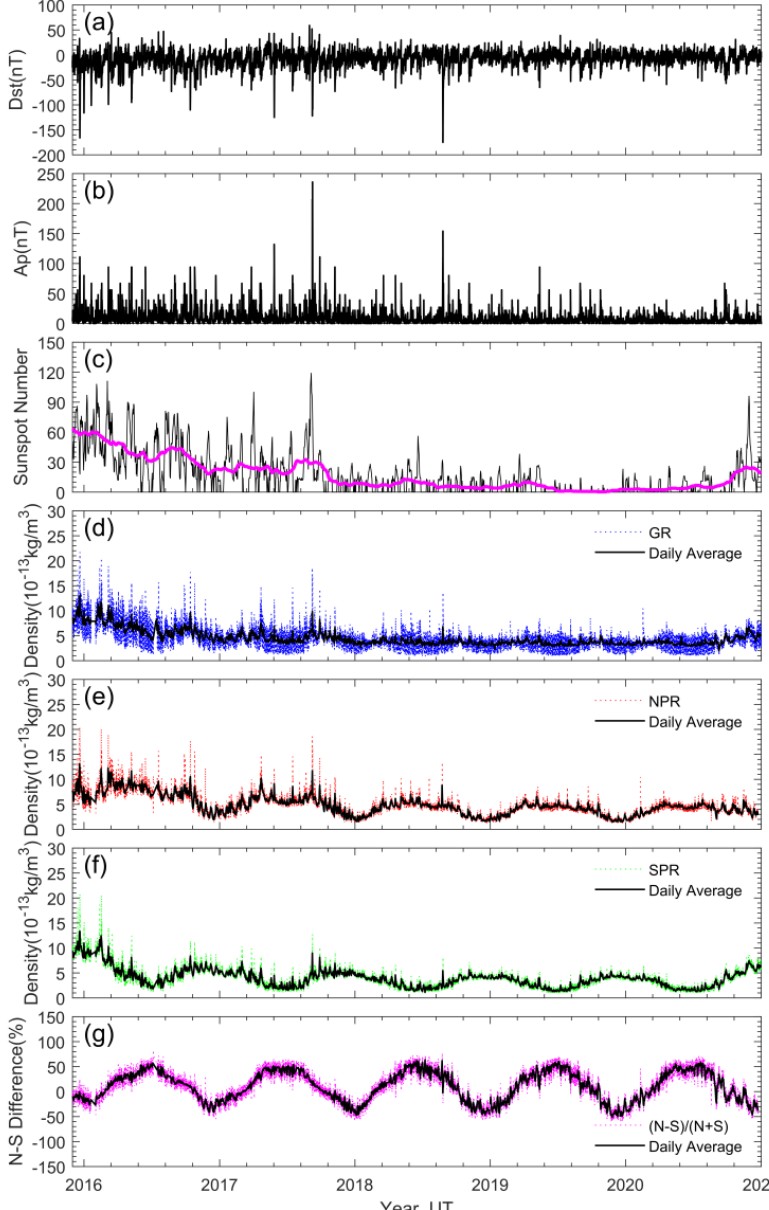

**Figure 7.** Geomagnetic index and density variations from December 2015 to December 2020: (**a**) Dst index, (**b**) Ap index, (**c**) sunspots—the magenta line is the monthly average value, (**d**) global region (GR) atmospheric density, (**e**) NPR atmospheric density, (**f**) SPR atmospheric density, (**g**) hemispherical density difference, where the black lines in (**c**–**e**) and (**f**)represent the corresponding daily averages.

To gain a clearer understanding of the seasonal variation characteristics of thermospheric density, we analyzed the relative difference of the monthly mean NPR, SPR, and GR densities from their annual mean values during the period from 2016 to 2020, as shown in Figure 8. In Figure 8b, the most obvious feature is the minimum value occurring in the summer months of June and July and the maximum in winter, which is expected. For the Northern Hemisphere, however, the density season characteristics are complicated. Except that the density peak of 2008 in June in summer, the peaks in other years were all in March or April in spring. In particular, the NPR density in 2017 shows two peaks, located in April and September respectively. This "double-peak" phenomenon is roughly the same as past studies [26,32]. Therefore, NPR and SPR densities in different hemispheres exhibit different distribution characteristics in terms of seasonality. Figure 8c shows that the thermospheric GR mean density reaches its maximum in the months of the spring and autumn equinoxes,

while its minimum value occurs in the summer, exhibiting similar features to the SPR density distribution. This seasonal asymmetry in thermosphere density is thought to result from interhemispheric circulation on a global scale [33].

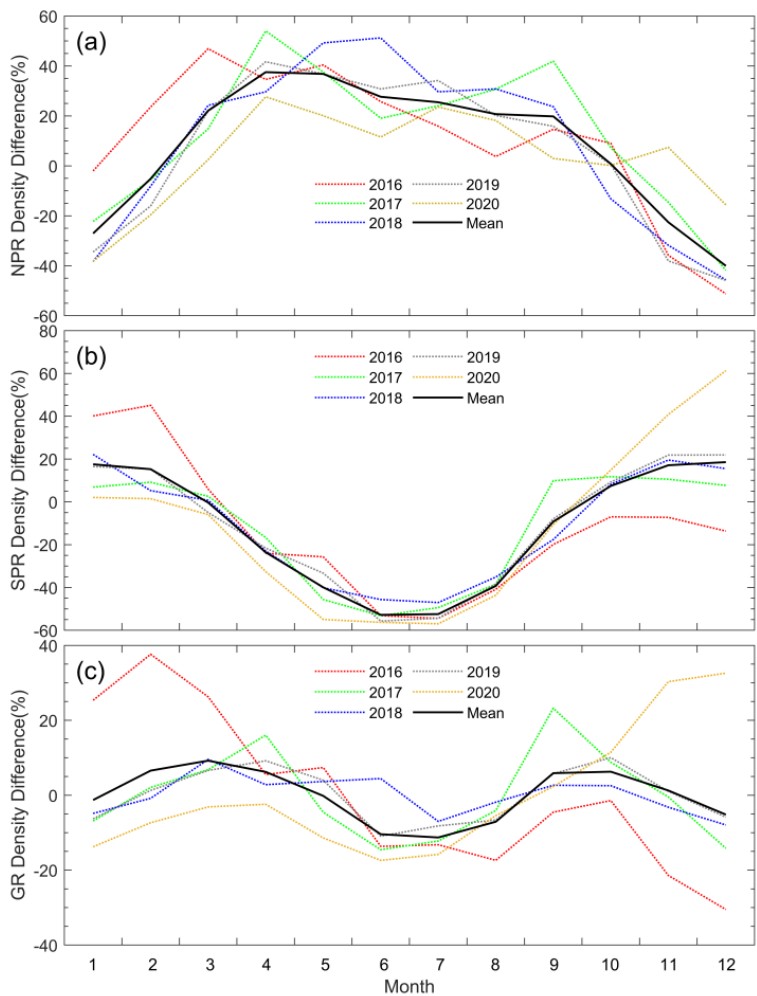

**Figure 8.** Seasonal variation in thermosphere density difference of (**a**) NPR, (**b**) SPR and (**c**) global region (GR) monthly averages relative to one year from 2016 to 2020.

## 4. Conclusions

In this study, the in situ atmosphere density data from the Atmosphere Density Detector (ADD) onboard the APOD satellite were used to analyze the asymmetry of atmospheric density in the northern and southern high-latitude regions of the thermosphere. Firstly, the geomagnetic response of atmospheric density in both hemispheres of the thermosphere during a dual magnetic storm event near the autumnal equinox on 8 September 2017 was studied. Then, the seasonal and annual solar activity characteristics of the atmospheric density differences between NPR and SPR in the thermosphere from December 2015 to December 2020 were explored over a five-year period under different geomagnetic activity conditions. The main results are summarized as follows.

1. For the magnetic storm event on 8 September 2017, the duskside SPR atmospheric density enhancement ratio is about 1.33–1.65 times that of NPR, showing a strong hemisphere density response asymmetry during the magnetic storm period. In the dawnside, this north–south response difference is smaller and also shows a dependence on LT differences. Energy injection in high-latitude regions leads to local atmospheric density enhancement and forms traveling atmospheric disturbance (TADs). These TADs can propagate to mid-low latitude regions and affect the global distri-

bution of thermospheric atmospheric density, with a propagation time in the order of hours. The relationship between the Dst and Ap indices and the hemisphere density enhancement ratio was quantitatively analyzed, and the fitting slope of the SPR relative density difference is higher than that of the NPR. This response asymmetry can be explained by SuperDARN plasma convection velocity. The plasma convection velocity of the SPR is significantly greater than that of the NPR, which indicates stronger joule heating caused by ion-neutral friction in the Southern Hemisphere.

2. Analysis of the long-term variation in atmospheric density asymmetry in the thermosphere high-latitude region of both hemispheres showed that it is influenced by solar activity, season, and different levels of geomagnetic disturbances. The thermospheric global (GR) atmospheric density decreases overall with decreasing sunspot numbers. The difference in atmospheric density between NPR and SPR has a clear annual periodicity. The distribution of NPR and SPR atmospheric density shows different seasonal characteristics. The NPR density peak is mainly in March or April. In particular, the "double-peak" phenomenon occurred in 2017, with peaks in March and September, respectively, while the largest feature of SPR atmospheric density is that its minimum value occurs in the summer months of June and July.

Some past studies have also shown that the interplanetary magnetic field and dipole tilt angle have obvious modulation of plasma convection, thereby affecting mass density hemispheric asymmetry. In addition, neutral atmospheres have tidal effects. The relationship between thermosphere density asymmetry and those will be explored in the future.

**Author Contributions:** Conceptualization ideas, J.A.; visualization, J.A.; writing—original draft preparation, J.A; validation, X.Z. (Xianguo Zhang), Y.L. and X.Z. (Xiaoliang Zheng); data provision, G.C.; writing—review and editing, C.X. and Z.Z. All authors have read and agreed to the published version of the manuscript.

**Funding:** This research was funded by the China Manned Space Program, grant Y59003AC40.

**Institutional Review Board Statement:** Not applicable.

**Informed Consent Statement:** Not applicable.

**Data Availability Statement:** The in situ density data of APOD were provided by the Beijing Aerospace Control Center [34]. The space environment indices such as solar wind came from NASA OMNIWeb data (https://cdaweb.gsfc.nasa.gov accessed on 21 March 2023). The sunspot number data are available at sidc.be/silso/datafiles (accessed on 1 April 2023). The SuperDARN data are from http://vt.superdarn.org/ (accessed on 9 April 2023).

**Acknowledgments:** We would like to express our heartfelt gratitude to the National Space Science Center of the Chinese Academy of Sciences for their full support of our research work. We appreciate JiaoJiao Zhang from National Space Science Center of the Chinese Academy of Sciences for her help in this paper. We also thank data sharing websites such as NASA for their support of our work. The authors acknowledge the use of SuperDARN data. SuperDARN is a collection of radars funded by national scientific funding agencies of Australia, Canada, China, France, Italy, Japan, Norway, South Africa, *United Kingdom and the United States of America*. In particular, we are sincerely grateful to the reviewers for their valuable and helpful comments.

**Conflicts of Interest:** The authors declare no conflict of interest.

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
