# Peer review of "Study on the Hemispheric Asymmetry of Thermospheric Density Based on In-Situ Measurements from APOD Satellite"

_atmosphere, doi:10.3390/atmos14040714_

Round 1
Reviewer 1 Report
In this manuscript, the authors used high spatiotemporal resolution data obtained by the atmospheric density detector carried by China's APOD satellite to study the hemispheric asymmetry of thermospheric density. They conducted both a case study and a statistical study. The found that there are the annual, seasonal, and magnetic storm response characteristics of the hemispheric asymmetry of thermospheric atmospheric density.
The paper has a high-quality. Nevertheless, I found several weaknesses in the manuscript. After these weaknesses are addressed, I would recommend publishing this paper.
Major suggestions:
Section 3.3:
1. What is the definition of tilt angle? Is it the angle between the dipole axis and the GSM z axis? Or is it the angle between dipole axis and earth rotation axis? I assume it is the angle between the dipole axis and the GSM z axis in this paper.
2. Besides annual variation, it should also have diurnal variation. Are you using a daily average tilt angle?
3. Tilt angle is strongly correlated with seasons, for example, -cos(Day of Year/365*2*pi). Thus, it is very hard to tell whether it is the tilt angle or the solar radiation that leads to the periodicity in N-S difference. Thus, I suggest removing this section, which will not impact the key points of this paper. Considering the huge amplitude of N-S difference, the solar radiation may be more important.
4. If you want to analyze the effect of tilt angle on N-S difference, I suggest investigating whether N-S difference has UT variation.
Minor suggestions:
Reorganize the introduction section. This long paragraph is hard to read. Instead of listing all related articles, it is better to summarize some common key points.
What is the definition of NPR and SPR? Above which latitude?
Line 169: The DST of the second storm is weaker than that of the first, so is it better to state that the effect of the second storm is stronger in the thermosphere or substorm activities?
Line 296: Not all years have two-peak structures. I think only 2017 has a clear two-peak structure. On the other hand, 2018 has only one peak in June. What is the statistical error of each point? If the statistical error is large, the two peaks may be artificial. I suggest using plateau instead of two peaks to describe this wide peak.
Author Response
Dear Reviewer:
First of all, we would like to express my sincere gratitude for your valuable comments on this paper. Here are some of your suggestions and my corresponding responses.
- Major suggestions
Section 3.3:
- What is the definition of tilt angle? Is it the angle between the dipole axis and the GSM z axis? Or is it the angle between dipole axis and earth rotation axis? I assume it is the angle between the dipole axis and the GSM z axis in this paper.
- Besides annual variation, it should also have diurnal variation. Are you using a daily average tilt angle?
- Tilt angle is strongly correlated with seasons, for example, -cos(Day of Year/365*2*pi). Thus, it is very hard to tell whether it is the tilt angle or the solar radiation that leads to the periodicity in N-S difference. Thus, I suggest removing this section, which will not impact the key points of this paper. Considering the huge amplitude of N-S difference, the solar radiation may be more important.
- If you want to analyze the effect of tilt angle on N-S difference, I suggest investigating whether N-S difference has UT variation.
Reply: As you said, there is a strong seasonal variation in dipole tilt angle, and it is difficult to explain that the N-S difference is caused by dipole tilt angle by listing the correlation coefficient of the in dipole tilt angle and N-S density difference separately. So I follow your advice and delete the part of the dipole angle.
- Minor suggestions
- Reorganize the introduction section. This long paragraph is hard to read. Instead of listing all related articles, it is better to summarize some common key points.
Reply: My Introduction is indeed a bit obscure. After my revision, my introduction mainly introduces the main input sources in the thermosphere including EUV radiation, particle deposition, and Joule heating, and discusses their effects on thermosphere density and composition, including seasonal and local time asymmetry.
- What is the definition of NPR and SPR? Above which latitude?
Reply: This is first explained in the revised version of Introduction
NPR : northern polar region that latitude above 60ºN;
SPR : southern polar region that latitude above 60ºS;
- Line 169: The DST of the second storm is weaker than that of the first, so is it better to state that the effect of the second storm is stronger in the thermosphere or substorm activities?
Reply: September 8, 2017 was a dual magnetic storm event caused by the CME, and as I said in my paper, since the first magnetic storm has not yet returned to the pre-disturbance state, the second magnetic storm has already begun, so the second magnetic storm may have a stronger impact on the thermosphere. Instead of just looking at the value of Dst. Maybe I just talked about the magnitude of the geomagnetic index earlier to make you misunderstood.
- Line 296: Not all years have two-peak structures. I think only 2017 has a clear two-peak structure. On the other hand, 2018 has only one peak in June. What is the statistical error of each point? If the statistical error is large, the two peaks may be artificial. I suggest using plateau instead of two peaks to describe this wide peak.
Reply: As Figure 8 shows, not all years have a “double-peak” structure, and my statement can indeed be a bit problematic. This section is mainly intended to describe the seasonal difference in density between the NPR and SPR. It is generally considered to have the minimum in the southern hemisphere and the maximum in the northern hemisphere on summer. From our results, the southern hemisphere is consistent. However, the maximum value in the Northern Hemisphere is mainly in the spring, and the “double-peak” structure also appeared in 2017, which is contrary to people's understanding, but previous studies also have seen similar phenomena.
We sincerely appreciate your valuable suggestions!
2023/4/8

Reviewer 2 Report
This paper first report the interhemispheric asymmetry of the mass density near 475 km observed by APOD in dawnside and duskside. However, when we are expecting the explanation of the asymmetry, they just provide the correlation analysis with Dst/Ap. They further provide the statistics of the interhemispheric asymmetry from 2015 to 2020. And then finally provide some surficial explanation. I think this paper needs major revision before it is suitable for publication.
Major comments: the paper now is a mixed soup. The author shall focus only one aspect. The interhemispheric asymmetry of the mass density during a equinox storm is pretty interesting. The author shall try their best to provide explanation on this phenomenon. Not just carrying out some unrelated work with this thing. I suggest the author collect other related data such as SUPER DARN plasma convection to explain why there is asymmetry in north and south polar region.
For the fitting with Dst/Ap, the resulted parameter is too low to prove they are corelated. Note that Dst is the index to reflect the response in equatorial region, while Ap is just index based on Kp to reflect the mid-latitude response. While here the author appears to spend most of their focus on polar region. I suggest the author remove the fitting, which is useless and did not provide any help in the explanation
For the statistic 2015-2020, I think this can be put in a separate paper as future work.
Line 73-74 here the author shall also cite some recent publications on the thermosphere composition. Note that ref 5 and 7 are not thermosphere composition, but thermosphere mass density, which is not suitable to appear near this sentence. The typical recent publications about thermosphere composition are as follows:
Cai, X., Wang, W., Lin, D., Eastes, R. W., Qian, L., Zhu, Q., et al. (2023). Investigation of the Southern Hemisphere mid-high latitude thermospheric ∑O/N2 responses to the Space-X storm. Journal of Geophysical Research: Space Physics, 128, e2022JA031002. https://doi.org/10.1029/2022JA031002 (and reference therein)
Yu, T., Wang, W., Ren, Z., Yue, J., Yue, X., & He, M. (2021). Middle-low latitude neutral composition and temperature responses to the 20 and 21 November 2003 superstorm from GUVI dayside Limb measurements. Journal of Geophysical Research: Space Physics, 126(8), e2020JA028427. https://doi.org/10.1029/2020ja028427
Line 115-147 here the author shall also describe the measurement error of the mass density
Line 150-169 this can be a separate sub-section with a title of ‘geomagnetic conditions during the storm on Sept 8, 2017
Then from line 174, the section shall be titled with something like ‘mass density behavior from Sept 7 to Sept 10, 2017’
Line 174-208 here the author shall also describe the density variations in low and mid-latitudes, not just high-latitudes
Line 227 It should be noted that
Line 229 start the new sentence from ‘This is because’
Line 229 remove ‘slightly’
Line 228 quiet period
Line 232-243 for the TAD, could the author give the phase speed and propagation direction??
Line 240 and continued to XXX
Line 242-243 this is useless, please delete it. It is complex, so what is the point??
For section 3.3, the author shall also discuss about the role of IMF By, since it is also crucial for the asymmetry.
Author Response
Dear Reviewer:
First of all, we would like to express my sincere gratitude for your valuable comments on this paper. Here are some of your suggestions and my corresponding responses.
- Major comment
- the paper now is a mixed soup. The author shall focus only one aspect.
Reply: My Introduction is indeed a bit obscure. After my revision, my introduction mainly introduces the main input sources in the thermosphere including EUV radiation, particle deposition, and Joule heating, and discusses their effects on thermosphere density and composition, and the resulting hemispheric asymmetry.
- The interhemispheric asymmetry of the mass density during a equinox storm is pretty interesting. The author shall try their best to provide explanation on this phenomenon. Not just carrying out some unrelated work with this thing. I suggest the author collect other related data such as SUPER DARN plasma convection to explain why there is asymmetry in north and south polar region.
Reply: It is true that I spent a great deal of article length describing the difference in density response between the northern and southern hemispheres caused by the magnetic storm of September 8, 2017, but did not explain why this asymmetry occurred. Based on your suggestions, SuperDarn plasma convection data was collected and visualized to obtain a convection map. The plasma convection velocity of SPR is significantly greater than that of NPR from Figure 6a and 6b. This indicates may stronger Joule heating caused by ion-neutral friction in the Southern Hemisphere, resulting in density response asymmetry in the NPR and SPR.
- Other comment
- For the fitting with Dst/Ap, the resulted parameter is too low to prove they are corelated. Note that Dst is the index to reflect the response in equatorial region, while Ap is just index based on Kp to reflect the mid-latitude response. While here the author appears to spend most of their focus on polar region. I suggest the author remove the fitting, which is useless and did not provide any help in the explanation
Reply: As you said, Dst/Ap describes the magnetic field at the equator/mid-latitude and may not be suitable for analyzing the correlation between Dst/Ap and density differences. This part is mainly to quantify the asymmetric response of the thermospheric NPR and SPR density whole magnetic storm.
- Line 73-74 here the author shall also cite some recent publications on the thermosphere composition. Note that ref 5 and 7 are not thermosphere composition, but thermosphere mass density, which is not suitable to appear near this sentence. The typical recent publications about thermosphere composition are as follows:
Reply: Thanks for your correction, these two articles are listed in References 12 and 13 respectively.
Cai, X., Wang, W., Lin, D., Eastes, R. W., Qian, L., Zhu, Q., et al. (2023). Investigation of the Southern Hemisphere mid-high latitude thermospheric ∑O/N2 responses to the Space-X storm. Journal of Geophysical Research: Space Physics, 128, e2022JA031002. https://doi.org/10.1029/2022JA031002 (and reference therein)
Yu, T., Wang, W., Ren, Z., Yue, J., Yue, X., & He, M. (2021). Middle-low latitude neutral composition and temperature responses to the 20 and 21 November 2003 superstorm from GUVI dayside Limb measurements. Journal of Geophysical Research: Space Physics, 126(8), e2020JA028427. https://doi.org/10.1029/2020ja028427
- Line 115-147 here the author shall also describe the measurement error of the mass density
Reply: The revised version adds brief detection elements of atmospheric density detectors and gives approximate density measurement errors: near 10%.
- Line 150-169 this can be a separate sub-section with a title of ‘geomagnetic conditions during the storm on Sept 8, 2017
- Then from line 174, the section shall be titled with something like ‘mass density behavior from Sept 7 to Sept 10, 2017’
- Line 174-208 here the author shall also describe the density variations in low and mid-latitudes, not just high-latitudes
- Line 227 It should be noted that
- Line 229 start the new sentence from ‘This is because’
- Line 229 remove ‘slightly’
- Line 228 quiet period
- Line 232-243 for the TAD, could the author give the phase speed and propagation direction??
- Line 240 and continued to XXX
- Line 242-243 this is useless, please delete it. It is complex, so what is the point??
Reply: I have responded to these comments together, and some grammar and titles have been modified with reference to your comments. Like Recommendation 6, the Revised version provides a brief description in these lines because of the analysis later in the analysis of the spread of TADs to low latitudes. For Recommendation 11, there is no ability to determine the phase velocity and propagation direction. Thanks to the suggestions and this direction will be studied later.
- For section 3.3, the author shall also discuss about the role of IMF By, since it is also crucial for the asymmetry.
Reply: Some past studies have also shown that the interplanetary magnetic field and dipole tilt angle have obvious modulation of plasma convection, thereby may affect the mass density hemispheric asymmetry. In addition, neutral atmospheres have tidal effects, the relationship between the thermosphere density asymmetry and them will be explored in the future.
We sincerely appreciate your valuable suggestions!
2023/4/8

Reviewer 3 Report
1. Please mention and interpret important numerical and statistical results of the paper in the abstract.
2. The introduction of the paper has been revised and similar researches of recent and more recent years should be mentioned.
3. The quality of the figures of the paper is low and cannot be read. Edit them. Must have a high resolution.
4. Interpret the numerical results in full. Please provide the interpretation of these results after the each figure.
Author Response
Dear Reviewer:
First of all, we would like to express my sincere gratitude for your valuable comments on this paper. Here are some of your suggestions and my corresponding responses.
- Please mention and interpret important numerical and statistical results of the paper in the abstract.
Reply: Based on your prompts, possible explanations of the phenomenon have been added to the revised version.
- The introduction of the paper has been revised and similar researches of recent and more recent years should be mentioned.
Reply: My Introduction is indeed a bit obscure. After my revision, my introduction mainly introduces the main input sources in the thermosphere including EUV radiation, particle deposition, and Joule heating, and discusses their effects on thermosphere density and composition, including seasonal and local time asymmetry. Similar recent studies are cited and presented in the Introduction.
- The quality of the figures of the paper is low and cannot be read. Edit them. Must have a high resolution.
Reply: The images in my paper are all high-resolution, and is still clear when word zoomed in to maximum(500%).
- Interpret the numerical results in full. Please provide the interpretation of these results after the each figure.
Reply: Based on your suggestions, we have made the necessary explanations for each image of the paper in the revised version.
We sincerely appreciate your valuable suggestions!
2023/4/8

Round 2
Reviewer 2 Report
Now the author has answered all my concerns and questions, the paper can be published after a minor revision. Although I still think there is no need to add the long term statistical study of the mass density, the author at least give the explanation of the asymmetry in density response and it is ok. I will provide my feedback based on the lines in the track changes
Lines 18-20 It should be 'the asymmetry response is smaller in the dawnside'
Line 64-66 here the asymmetry is actually not related to the 11-year solar irradiation variation. Please correct into: Due to the different solar irradiation received in seasons, thermosphere mass density and composition can also show corresponding seasonal variations
Line 69 GUVI observations to show
Line 76 remove 'atmosphere'
Line 83 remove 'statistical studies'
Line 84 what do you mean by highly-structured
Line 90 column density ratio of Oxygen to Nitrogeon (∑O/N2)
Line 369 please give certain value range since you are able to do so,
Line 479 the largest
Also, please give the description of the convection before and after, or put other UTs convection fig in supporting information
Author Response
Dear Reviewer:
Thank you very much for your comments on the revision. Here are some of your suggestions and my corresponding responses.
- Comments and Suggestions
- Lines 18-20 It should be 'the asymmetry response is smaller in the dawnside'
- Line 64-66 here the asymmetry is actually not related to the 11-year solar irradiation variation. Please correct into: Due to the different solar irradiation received in seasons, thermosphere mass density and composition can also show corresponding seasonal variations
- Line 69 GUVI observations to show
- Line 76 remove 'atmosphere'
- Line 83 remove 'statistical studies'
- Line 90 column density ratio of Oxygen to Nitrogen (∑O/N2)
- Line 479 the largest
Reply: I have revised these comments according to your suggestions
- Line 84 what do you mean by highly-structured
Reply: “highly-structured”: The peak density is located in the cusp, showing this structured distribution, which is related to particle precipitation and field aligned current heating
From Reference 8
- Line 369 please give certain value range since you are able to do so,
Reply: Added description of plasma convection values for the northern and southern hemispheres.
- Also, please give the description of the convection before and after, or put other UTs convection fig in supporting information
Reply: Indeed, it is interesting to understand the development of plasma convection in the northern and southern hemispheres throughout the magnetic storm, but due to space limitations, I have added a description of plasma convection in the northern and southern hemispheres during the main phase of the second magnetic storm to the text.
I add here a 2-min convection pattern plot before and after the UT time analyzed for the paper. It is clear that plasma convection velocities in the high latitudes of the southern hemisphere are greater than in the northern hemisphere.
Figure 1. The two-min SuperDARN convection pattern
We sincerely thank you for reading this paper and making suggestions so carefully!
2023/4/10